# Spanish Cross-Cultural Adaptation and Validation of Neck Bournemouth Questionnaire (NBQ) for Neck Pain Patients

**DOI:** 10.3390/healthcare11131926

**Published:** 2023-07-03

**Authors:** Juan José Pérez-García, Guadalupe Molina-Torres, María Isabel Ventura-Miranda, Irene Sandoval-Hernández, María Dolores Ruiz-Fernández, Jesús Martínez-Cal, Manuel Gonzalez-Sanchez

**Affiliations:** 1Department of Nursing, Physiotherapy and Medicine, Faculty of Health Sciences, University of Almería, 04120 Almeria, Spain; jpg122@inlumine.ual.es (J.J.P.-G.); guada.lupe@ual.es (G.M.-T.); mvm737@ual.es (M.I.V.-M.); mrf757@ual.es (M.D.R.-F.); 2Department of Physical Therapy, Faculty of Health Sciences, University of Granada-Campus of Melilla, C/Santander, 1, 52005 Melilla, Spain; isandoval@ugr.es; 3Facultad de Ciencias de la Salud, Universidad Autónoma de Chile, 7500912 Providencia, Chile; 4Department of Physiotherapy, University of Malaga, 29071 Málaga, Spain; mgsa23@uma.es; 5Institute of Biomedical Research of Malaga (IBIMA), 29010 Malaga, Spain

**Keywords:** cross-cultural adaptation, questionnaires, neck pain, Neck Bournemouth Questionnaire, surveys, validation

## Abstract

Background: Neck pain is highly prevalent and one of the most common musculoskeletal conditions. Instruments that measure the factors involved in neck pain accurately are needed for clinical assessment. Patient-reported outcome measures (PROMs) are reliable, cost-effective, and specific tools for the assessment of musculoskeletal problems at different moments. The Neck Bournemouth Questionnaire (NBQ) assesses pain, function, disability, and psychological and social variables in patients with cervical pathologies. The aim of this study was to perform an adaptation and validation into Spanish of the NBQ (NBQ-Sp). Methods: A cross-sectional, observational study was carried out through translation, adaptation, and validation. A total of 129 patients with neck pain, of Spanish nationality, and over 18 years of age were included. Results: The NBQ-Sp showed excellent internal consistency, with Cronbach’s α of 0.897, test–retest reliability with interclass correlation coefficient (ICC) of 0.866, and standard error of measurement (SEM) and minimum detectable change (MDC) values were 1.302 and 3.038, respectively. A Kaiser–Meyer–Olkin value of 0.857 was obtained, and Bartlett’s test yielded *p* < 0.001, finding one factor in the factor analysis. Conclusion: The NBQ-Sp has proven to be a valid and reliable tool for clinicians and researchers to measure neck pain in the Spanish population.

## 1. Introduction

Musculoskeletal pain is a very common problem in society and is one of the main causes of disability in the working population, both men and women, causing high health costs in developed countries [1,2].

Neck pain (NP) is one of the most common musculoskeletal conditions, defined as an “unpleasant sensory and emotional experience” in the cervical region [3], with a global prevalence of 288.7 million and an age-standardized point prevalence per 100,000 population of 3551.1 and incidence per 100,000 population of 806.6 according to the Global Burden of Disease study [2,4,5,6], and it is considered a top-level public health issue [4,5,7]. Prevalence patterns by sex and age show that NP is more prevalent in women than in men in all age-groups, although it peaks in middle age [2]. By age range, it is more prevalent in women aged 45–54 years and in men aged 45–49 years [2,5]. It has a high health-care cost due to absenteeism from work. In 2016, NP, combined with low-back pain, was one of the conditions with the highest health-care spending in the United States, with a total of $134.5 billion [7,8].

Imaging tests have proven to be ineffective for the diagnosis of NP, since there may not necessarily be radiological evidence of tissue damage for pain to exist [9,10]. NP must be understood not only from a physical point of view but also from a biopsychosocial and individual point of view. Therefore, tools are needed to measure all these factors accurately and reliably in order to be able to carry out a correct clinical follow-up and to be able to measure the efficacy of the treatments [4,5,10]. Patient-reported outcome measures (PROMs) are an excellent resource for patients’ views on their symptoms, activities of daily living (ADL) performance, functional status, and health-related quality of life at different moments in time. PROMs are useful for clinicians and researchers to interpret the results of patients’ evolution and changes in their symptoms, capacity, and function [10].

The Neck Bournemouth Questionnaire (NBQ) [11] was validated by Bolton et al. in 2002 on the assumption that NP, like low-back pain, is explained by a model of musculoskeletal pain [11]. The NBQ consists of seven items with numerical scales from 0 to 10 to assess pain from a biopsychosocial point of view, in addition to disability, which helps us to have a better understanding of NP. The brevity of the questionnaire helps for better application in clinical and research settings [11,12,13].

The most widely used and widely used NP-specific questionnaire among researchers is the Neck Disability Index (NDI) [14]. While this focuses on determining the degree of disability and pain, the NBQ aims to go a step further by taking a psychological approach to NP, in addition to measuring disability and degree of pain. It was conceived from a biopsychosocial model of pain, as knowledge of these biological and psychological factors have proven to be effective in understanding self-reported pain [11].

Questionnaires must be translated and culturally adapted to the environment in which they are to be used. The psychometric properties must then be assessed to ensure that the tool has exactly the same characteristics, validity, and reliability as the original version [11]. The NBQ was originally created in English and has been translated and validated in different languages, among which Spanish is not included, even though it is among the five most widely spoken languages and is considered the second-most spoken mother tongue in the world, according to the United Nations (UN) [15]. A Spanish validation of the NBQ (NBQ-Sp) would provide an objective assessment resource and would allow the planning of therapeutic strategies for the prevention of NP and for its treatment once it is established.

The aim of this study was to adapt and validate a Spanish version of the Neck Bournemouth Questionnaire (NBQ-Sp), an extensively utilized clinical evaluation instrument for assessing disability, pain intensity, and quality of life in neck pain patients, with a biopsychosocial approach.

## 2. Methodology

### 2.1. Design

A cross-sectional, observational study was conducted, which consisted of two phases: (A) translation and cross-cultural adaptation of the NBQ into Spanish (NBQ-Sp) and (B) validation of the NBQ-Sp.

### 2.2. Study Setting

Participants were recruited from physiotherapy clinics, traffic accident units, and students from the University of Almería, Spain from October 2022 to March 2023.

### 2.3. Study Population

Patients with nonspecific neck pain were included in the study. Inclusion criteria were: (1) patients with acute (up to 6 weeks), subacute (between 6 and 12), or chronic neck pain (more than 12 weeks) [4] and (2) Spanish-speaking patients over 18 years of age. Exclusion criteria were: (1) patients who did not properly complete and return the questionnaire and (2) those who—due to cognitive impairment and/or lack of understanding of the questions—were unable to answer the forms. The study sample was established by using a convenience sampling method based on availability and accessibility. The sample size was determined following the recommendations of Kline et al. [16], who recommended a sample of 10–20 participants per item. The NBQ has 7 items; therefore, a sample of 70–140 participants is adequate.

### 2.4. Ethics Considerations

All ethics considerations were taken into account at all times. This study was conducted in accordance with the ethical principles for medical research involving human subjects according to the Declaration of Helsinki. The data were used in accordance with the Organic Law 3/2018 of 5 December on Personal Data Protection and guarantee of digital rights. All participants signed informed consent forms to take part in the study. This study was approved by the Ethics and Research Commission of the Department of Nursing, Physiotherapy and Medicine of the University of Almería (registration EFM 215/2022).

### 2.5. Neck Bournemouth Questionnaire (NBQ)

The NBQ was developed by Bolton and Humphreys in 2002 because of the need for a measure to assess several health domains, such as pain, function, disability, and psychological and social aspects of patients with cervical pathologies. The NBQ consists of 7 items: (1) pain intensity, (2) performance in activities of daily living, (3) social activities, (4) anxiety, (5) depression, (6) how work affects their pain, and (7) with patients’ control over their pain. Each item is given a score from 0 to 10 on a numerical scale, giving a maximum score of 70 points. It has a Cronbach’s α of 0.9 [11].

### 2.6. Translation and Adaptation

For the translation and cross-cultural adaptation, the recommendations of the International Test Commission Guidelines for test translation and adaptation were followed [17] to ensure terminological and conceptual equivalence, in the questions that make up the Spanish version of the NBQ. For the translation from English of the Spanish version of the NBQ, a 5-step protocol was followed by two native Spanish speakers. Both translations were independent and the translators were blinded. In the next phase, the two independently translated versions of the NBQ were compared and pooled to reach an agreement on how to elaborate the draft version. Discrepancies between the two independent translators were resolved by a third reviewer. Next, a back-translation was carried out in which two native English translators independently translated the NBQ-Sp to check that the items in this version were a faithful reflection of the construct to be measured by the original version of the NBQ. Once the final version of the NBQ-Sp had been obtained, a pilot study was carried out with a sample of 30 participants, with the aim of identifying possible questions that were difficult for participants to understand, thus influencing their responses to the test. After the pilot test, the psychometric properties of the NBQ-Sp were analyzed with data obtained from a sample of patients suitable for this process. Figure 1 shows in schematic form the steps that were followed for the cross-cultural adaptation of the NBQ-Sp.

### 2.7. Questionnaires for Construct Validity

#### 2.7.1. Quality of Life SF-12

The SF-12 is composed of a set of 12 items on health-related quality of life, showing two reduced scores, the physical component status and the mental component status, on a scale from 0 to 100. It also presents eight additional domains (physical functioning, role physical, bodily pain, general health, vitality, social functioning, role—emotional, and mental health). Both the additional domains and the summary scores are calculated using algorithms where each item response has an individual weight in the total score. Higher scores indicate better health-related quality of life [18].

#### 2.7.2. EuroQoL 5-D/VAS

The EuroQol-5D is a questionnaire to measure people’s quality of life. It is composed of 5 domains (mobility, self-care, regular activities, pain/discomfort, and anxiety/depression) divided into three levels of severity (no problems, some problems or moderate problems and severe problems). This system also includes a visual analogue scale (EQ-5D VAS) defined by a 10 cm vertical scale, with each end representing extreme expressions of self-perceived health status ranging from 0 (worst health) to 100 (best health). It has a Cronbach’s α of 0.53 [19].

#### 2.7.3. Neck Disability Index (NDI)

The NDI is a 10-item tool developed in 1991 that assesses pain and disability in patients presenting with neck pain. The scale addresses issues such as pain intensity, self-care, lifting, reading, headaches, concentration, work, driving, sleep, and leisure. Each item is rated from 0 to 5. It is the most widely used neck pain scale in the largest number of populations and the most frequently validated. The NDI establishes a categorization of disability according to the test score: 0 to 4 no disability, 5 to 14 mild disability, 15 to 24 moderate disability, 25 to 34 severe disability and more than 34 total disability. It has a Cronbach’s α of 0.9 [20].

### 2.8. Data Collection

The data were collected through a digital self-reported questionnaire created using the Google Forms tool, which was sent via WhatsApp and/or email link to selected patients with cervical pain. The decision to conduct the questionnaire online was made with the intention of avoiding biases and not influencing the participants’ responses. Two questionnaires were sent, with a two-week difference between measurements. The questionnaires consisted of sociodemographic variables, the Spanish version of the NBQ (NBQ-Sp), the SF-12 questionnaire, the EuroQol-5D/VAS, and the Neck Disability Index (NDI).

### 2.9. Data Analysis

A descriptive analysis was performed for the sociodemographic variables, as well as for all the assessment measures included, where the mean and standard deviation were calculated. The Kolmogorov–Smirnov test was used to analyze the distribution and normality of the sample. Cronbach’s α coefficients were calculated to analyze the internal consistency of the measures. Cronbach’s α values were classified according to the following scale: Cronbach’s α ≤ 0.40 poor, 0.60 > Cronbach’s α > 0.40 moderate, 0.80 > Cronbach’s α ≥ 0.60 good, and Cronbach’s α ≥ 0.80 excellent [21]. Item responses were analyzed using the interclass correlation coefficient (ICC). For comparison of parametric variables, Student’s t-test, and the Wilcoxon test for nonparametric variables were used. The floor–ceiling effect was measured, being considered present if >15% of the participants obtained the lowest (floor) or highest score (ceiling) [22].

Construct validity was obtained by exploratory factor analysis (EFA) and subsequent confirmatory factor analysis (CFA). EFA was determined by Bartlett’s test and the Kaiser–Meyer–Olkin (KMO) measure. The KMO has a statistical range from 0 to 1, with values closer to 1 being the most appropriate and the better the correlation, although values >0.5 are considered adequate for EFA. Bartlett’s test results must be <0.05 for EFA to be performed [23].

Maximum likelihood extraction (MLE) was used to analyze structure and construct validity. Three requirements must be met: (1) explain >10% of the variance, (2) have an eigenvalue >1.0, and (3) have an inflection point in the scatterplot. In order to carry out the MLE, there must be a sample of at least 10 participants per item; the NBQ consists of 7 items, so at least 70 participants are needed [24].

To calculate the standard error of measurement (SEM), the formula SEM = s√1 − r; was used, where “s” refers to the standard deviation (SD) of the scores of the two measurements taken and “r” is the reliability coefficient of the Pearson test correlations between the test values and the retest.

To measure sensitivity or measurement error, the minimum detectable change 90 (MDC90) was calculated following the analysis described by Stratford32, using the following formula: *MDC*90 = *SEM* × √2 × 1.65.

The criterion validity was obtained by means of Pearson correlations between the NBQ-Sp and the Spanish versions of the questionnaires SF-12 [18], EuroQol 5-D [25] and NDI [20]. Pearson correlations were scored according to the following parameters: r ≤ 0.49 (poor), 0.50 ≤ r ≤ 0.74 (moderate), r ≥ 0.75 (strong) [26]. The statistical analysis of the study was carried out using the SPSS (V.28.0) statistical processing program.

## 3. Results

The items of the translated and adapted version of the NBQ into Spanish (NBQ-Sp) are shown in Table 1, and the full questionnaire is available in Appendix A. A total of 129 people participated in the study, of whom 96 were men and 33 women, with a mean age of 35.32 years and standard deviation (SD) of 13.47 years. In terms of educational level, 57.5% of the participants had higher education, such as a university degree, master’s degree or doctorate (PhD), 69.8% of the participants had chronic NP (more than 3 months duration), 30.2% of the participants had acute NP (up to 6 weeks duration) and subacute NP (6 to 12 weeks duration) [4], 74.4% of the participants had no congenital disease, and 69.8% of the participants would need to take medication to control NP symptoms. Table 2 shows the sociodemographic characteristics of the participants.

Table 3 shows the minimum, maximum, mean and standard deviations of the participants’ age data, and those provided by the measurement instruments used in this study, which were: SF-12, EuroQol (VAS and 5D) and NDI questionnaires. In the evaluation of the ceiling and floor effect, it was found that only one participant reached the maximum score (0.0078%) in his answers to the questionnaire and that none of the participants reached the minimum score. With these values, it cannot be said that there is a ceiling and floor effect in the NBQ-Sp.

For construct validity, the EFA was carried out. The KMO statistic was 0.857 and Bartlett’s test yielded *p* < 0.001 (*p* < 0.05), so the requirements were met (see Table 4). The EFA found a single factor with an eigenvalue above 1 and explaining 62.834 of the variance encompassing the seven items of the questionnaire (see Table 5). As only one factor was found, it was not possible to rotate these components using the Varimax method, and therefore the CFA could not be carried out. The communalities ranged from 0.681 to 0.767. The EFA scree plot is shown in Figure 2, with an inflection point at item 2, so the model has only one factor.

Table 6 shows the results of Cronbach’s α and reliability of the questionnaire: The overall Cronbach’s α of the test was 0.897, giving excellent internal consistency. Test–retest reliability was obtained from the analysis of the results of two NBQ-Sp measurements taken by the same participants in a 2-week time interval between attempts, resulting in an overall interclass correlation coefficient (ICC) of 0.866 (ICC 95%: 0.791–0.933, *p* < 0.05). The SEM and MDC90 scores were 1.302 and 3.038, respectively.

Correlations between the NBQ-Sp total score and the measurement instruments used were calculated to analyze criterion validity. The NBQ-Sp showed a strong correlation (≥0.75) with all subscales of the SF-12 questionnaire. Correlation levels ranged from r = 0.901 (mental health) to r = 0.978 (physical component state). A moderate correlation (0.50 ≤ r ≤ 0.74) was found with the total score of the NDI questionnaire, being r = 0.657, a poor (r ≤ 0.49) and moderate correlation was found with the NDI domains, with correlation ranges between r = 0.437 (personal care) and r = 0.565 (pain intensity). The correlation with the EuroQol 5D and VAS questionnaire was moderate, with values of r = −0.771 and r = −0.564, respectively. All these correlations are shown in more depth in Table 7.

## 4. Discussion

Based on the aim of the study, the psychometric properties were evaluated in a sample of Spanish patients with neck pain, and the NBQ-Sp was found to be a valid and reliable resource for use in the Spanish population, both for clinical assessment and in the research field.

### 4.1. Translation and Cross-Cultural Adaptation

The results of this study suggest that the questionnaire has been understood by the participants, thus interpreting that a correct translation and cross-cultural adaptation of the original English questionnaire into Spanish was carried out. The recommendations of the available literature were followed in order to favor the comparison of the results of this questionnaire with those obtained in the different language versions, such as the German [27], French [28], Italian [29], Dutch [30], Brazilian [12], Chinese [31], Persian [32] and Turkish I [33], II [34] y II [35] versions.

### 4.2. Construct Validity

The exploratory factor analysis (EFA) identified a single factor that had an eigenvalue >1, explained 62.834 of the variance, and had an inflection point at component 2 in the scree plot, meeting the requirements of MLE [24]. The authors of the original version [11] did not perform a factor analysis and therefore it cannot be compared, although the authors of the Italian [29], Turkish II [34] and III [35] versions did. The Italian version [29] had two factors with eigenvalues >1 and explaining 56.6% and 12.6% of the variance, respectively, for each factor, thus having one more factor in the analysis conducted in this study. However, the Turkish II [34] y III [35] versions, as in this study, also had a single factor, with variances of 59.08% for the Turkish I version [34] y and 58.23% for the Turkish III version [35].

A possible explanation for the fact that there are studies with one factor and others, such as the Italian version [29], with two factors, may be sociocultural differences between the two population groups. These differences could determine the weights of the different items in each of the factors, which would condition this difference. Even within the same country and culture, it is sometimes necessary to carry out validation analyses in different population groups to be able to evaluate possible differences between groups, as in the case of the Turkish validations I [33], II [34] and III [35], which, although a population from the same country was used, showed differences in results.

### 4.3. Internal Consistency and Reliability Test–Retest

The NBQ-Sp demonstrated excellent internal consistency with an overall Cronbach’s α value of 0.897 if we compare these results with those of the original version [11] which showed a Cronbach’s α of approximately 0.9, in accordance with the results of this study. For the different versions of the NBQ, we found that the Italian version [29] obtained a score of 0.89, a similar result to this study. The Chinese [31], Persian [32], Dutch [30] and Turkish II [34] versions obtained values of 0.87, very similar to those of the NBQ-Sp. On the other hand, other versions obtained a higher Cronbach’s α, such as the Brazilian [12] and Turkish I [33] versions, with 0.98, or lower Cronbach’s α, such as the German version [27], which obtained scores of 0.79–0.80.

When we compare the test–retest reliability values, we find that the results of this study show a total ICC of 0.866, which is adequate, but lower than the results obtained by the different versions of the NBQ. which showed an ICC of 0.91, which—although similar to those obtained in this study—is still below the other versions: 0.99 in general, as is the case of the German version [27]. In contrast, the total CCI value found in the NBQ-Sp is higher than that shown by the original version [11]: the authors indicated an ICC value of 0.65.

### 4.4. Measurement Error

The NBQ-Sp showed an SEM value of 1.302 and an MDC90 of 3.038. The SEM data indicate that an individual’s actual score on a test would be expected to vary by about 1302 points around their observed score, and the MDC90 data indicates that a change of at least 3038 units in the score of a questionnaire measurement is needed to be 90% sure that the observed change is not due to measurement error or chance. The Dutch version [30] was the only version to indicate SEM data and the Turkish version III the only one to indicate MDC data. The Dutch version [30] obtained an SEM value of 3.67, and the Turkish version III [35] provided MDC data of 20.31. The original version [11] did not report SEM or MDC data.

### 4.5. Criterion Validity

The questionnaires used for criterion validity were the SF-12, NDI, EuroQol 5D and VAS. The NBQ does not have specific dimensions or subscales, so the correlations were made taking into account the total score of the questionnaire. The highest correlations were found with the SF-12 questionnaire, although it also correlated adequately with the EuroQol 5D and VAS, as well as with the NDI. When comparing the correlations of the NBQ-Sp with those of the different versions, only the correlations with the NDI could be taken into account, which has been used in most of the versions and by the original version [11] as well. The NBQ-Sp correlated with the NDI with a value of r = 0.657, similar to that indicated by the French version [28], with r = 0.67, or to the original version [11], which obtained a value of r = 0.51 (pre) and r = 0.71 (post), although it was lower than other versions, such as the Dutch version [36], with r = 0.82, and higher than other versions, such as the Turkish version II [34], with r = 0.32. In this sense, we can say that the NBQ-Sp correlates similarly with the NDI to the rest of the versions and the original version. For future adaptations and validations of PROM, the questionnaires used for the criterion validity should be standardized to facilitate comparisons between the different versions.

### 4.6. Implications for Future Research and Clinical Administrations

It is essential that resources are available to assess and monitor different conditions. Therefore, the tools used should be developed according to best practice and supported by available evidence [4,10]. For this purpose, the NBQ was designed in English, and its Spanish version (NBQ-Sp) has proven to be a valid and reliable tool to assess cervical disorders in the Spanish population.

The NBQ-Sp has been developed using current recommendations from the scientific literature and has proven to be a useful tool for the assessment and follow-up of patients with cervical disorders. However, it is important to emphasize that during the conduct of this study, it was identified that some versions of the NBQ have not had analysis of some psychometric characteristics carried out, so future studies are needed to complete the validation process of these tools and ensure their reliability and validity in the population of Spain.

### 4.7. Strengths and Weaknesses

The translation and cross-cultural adaptation of the NBQ into Spanish (NBQ-Sp), as well as its validation carried out in this study, has allowed the use of this questionnaire in Spanish, the second-most spoken language in the world. The results of this study have been obtained from the analysis of the scores of an adequate sample, according to the recommendations of the literature. A minimum of 70 participants was required, as it is a seven-item questionnaire. This study was carried out with a total of 129 participants, a figure that meets this requirement and is in line with the samples used with the different versions and the original English version. It is a questionnaire that is easy to understand for the population and quick to administer in both the clinical and research fields, which facilitates its use by professionals.

On the other hand, the following weaknesses should be noted. Unlike other versions, the NBQ-Sp has not analyzed any longitudinal psychometric properties, such as responsiveness or sensitivity to change. It was not possible to rotate the components using the Varimax method, as only one factor was found in the factor analysis, so confirmatory factor analysis could not be performed either. Future validation studies should take into account the weaknesses of this study in order to improve the methodological and evidence quality of future versions of NBQ.

## 5. Conclusions

The Spanish version of the NBQ (NBQ-Sp) has proven to be a valid and reliable resource for the self-assessment of neck pain and understanding pain and disability from a biological, psychological and social context of each person. Its easy comprehension and speed of use makes it an ideal tool for clinicians and researchers to assess and monitor neck pain in the Spanish-speaking population.

## Figures and Tables

**Figure 1 healthcare-11-01926-f001:**
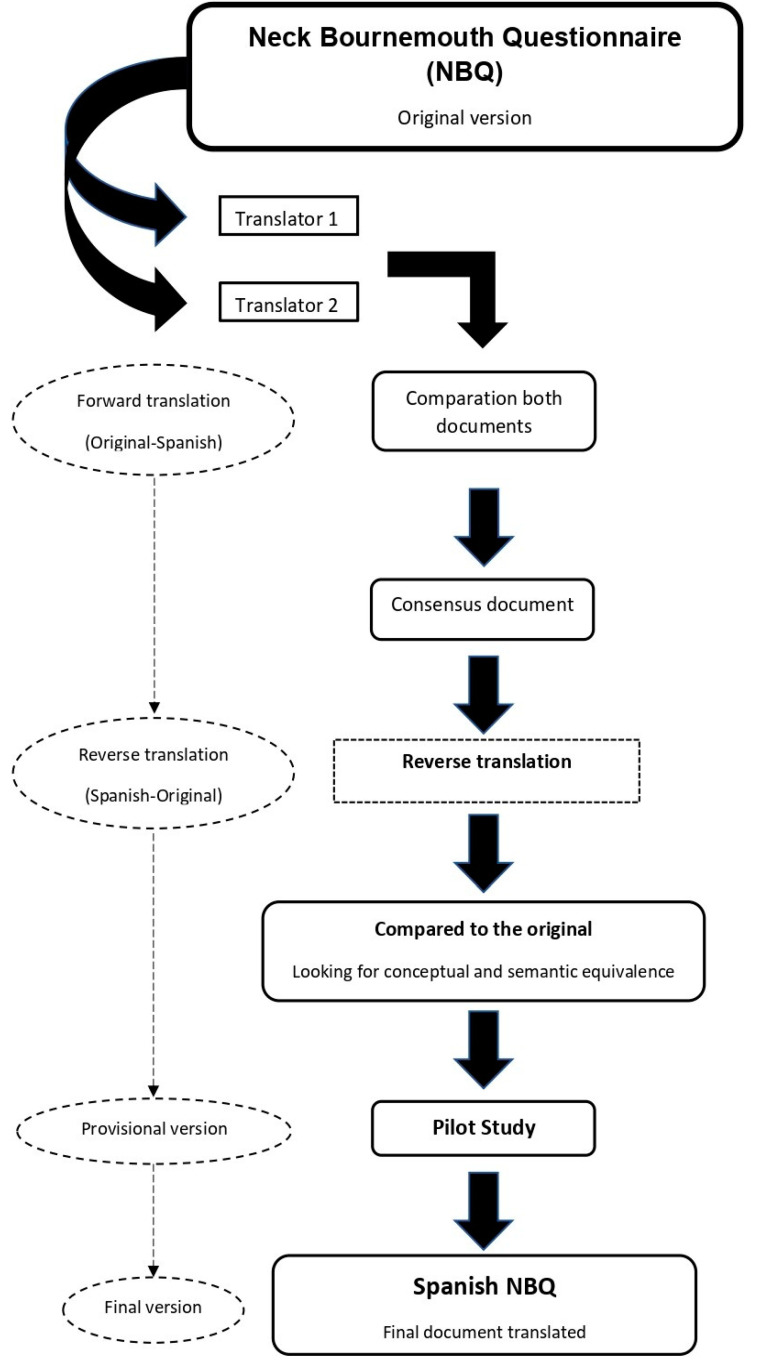
Flowchart of the cross-cultural adaptation process of the NBQ-Sp.

**Figure 2 healthcare-11-01926-f002:**
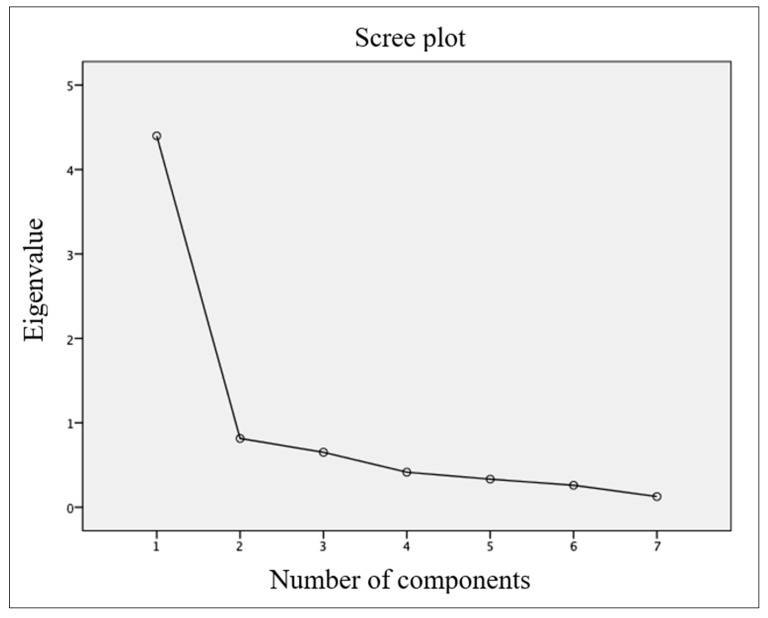
Scree plot of NBQ-Sp.

**Table 1 healthcare-11-01926-t001:** Items of Spanish version of the NBQ (NBQ-Sp).

Item 1	Durante la última semana, ¿cómo calificarías de media tu dolor de cuello?
Item 2	Durante la última semana, ¿en qué medida te ha molestado el dolor de cuello en tus actividades cotidianas (tareas domésticas, lavarse, vestirse, levantar peso, leer, conducir)?
Item 3	Durante la última semana, ¿en qué medida ha interferido el dolor de cuello en tu capacidad para participar en actividades recreativas, sociales y familiares?
Item 4	Durante la última semana, ¿hasta qué punto te has sentido ansioso/a (tenso/a, irritable, con dificultad para concentrarte/relajarte)?
Item 5	Durante la última semana, ¿cómo de deprimido (abatido, triste, desanimado, pesimista, infeliz) te has sentido?
Item 6	Durante la última semana, ¿has sentido que tu trabajo (tanto dentro como fuera de casa) ha afectado (o afectaría) a tu dolor de cuello?
Item 7	Durante la última semana, ¿hasta qué punto has podido controlar (reducir/ayudar) tu dolor de cuello por ti mismo?

**Table 2 healthcare-11-01926-t002:** Sociodemographic characteristics of participants.

		Age					Frequency	Percentage	Accumulated Percentage
		<29	30–39	40–49	50–59	>60			
Gender	Man	15	6	8	2	2	96	74.4	74.4
	Woman	49	12	16	11	8	33	25.6	100.0
Educational level	Basic Education	7	5	7	5	2	26	20.2	20.2
	Vocational training	13	3	7	3	3	29	22.5	42.6
	University studies	28	3	5	5	5	46	35.7	78.3
	Master’s Degree	15	7	4	-	-	26	20.2	98.4
	Doctorate	1	-	1	-	-	2	1.6	100.0
Duration of neck pain	Less than 3 months	15	6	11	2	5	39	30.2	30.2
	3 to 6 months	11	5	3	3	2	24	18.6	48.8
	From 6 to 12 months	5	-	-	-	-	5	3.9	52.7
	More than 1 year	33	7	10	8	3	61	47.3	100.0
Congenital disease	No	48	14	22	7	5	96	74.4	74.4
	Yes	16	4	2	6	5	33	25.6	100.0
Medication	No	51	12	18	5	4	90	69.8	69.8
	Yes	13	6	6	8	6	39	30.2	100.0

**Table 3 healthcare-11-01926-t003:** Descriptive statistics of the measuring instruments used in this study.

		Minimum	Maximum	Mean	Standard Deviation
Age		18	72	35.32	13.47
NBQ	Intensity of pain	0.00	10	5.63	2.194
	Activities of daily living	0.00	10	4.65	2.606
	Social activities	0.00	10	4.13	2.805
	Anxiety	0.00	10	5.74	2.841
	Depression	0.00	10	4.79	3.084
	Pain at work	0.00	10	5.88	2.736
	Self-management of pain	0.00	10	4.91	2.460
	Total	3.00	70	35.72	14.789
SF-12	Physical Function	22.11	67.16	47.87	11.85
	Role Physical	20.32	62.91	39.40	13.95
	Bodily Pain	16.68	63.90	44.75	12.23
	General Health	18.87	64.61	45.88	11.74
	Vitality	27.62	67.88	52.86	10.78
	Social Functioning	16.18	65.70	49.57	12.33
	Role Emotional	11.35	68.81	37.04	16.66
	Mental Health	27.97	65.73	48.41	11.57
	Physical Component State	17.43	64.84	45.89	12.76
	Mental Component state	29.21	75.48	54.32	9.46
EuroQol	VAS	10	100	66.00	1.87
	5D	0.00	1.00	0.73	0.20
NDI	Pain Intensity	0.00	5.00	1.97	1.068
	Personal care	0.00	5.00	0.45	0.901
	Lifting weights	0.00	5.00	1.18	1.302
	Reading	0.00	4.00	1.61	1.127
	Neck pain	0.00	5.00	2.09	1.444
	Concentrate on something	0.00	5.00	1.57	1.198
	Work and usual activities	0.00	5.00	1.26	1.314
	Driving	0.00	5.00	1.22	1.226
	Sleep	0.00	4.00	1.25	1.146
	Free time activities	0.00	5.00	1.32	1.262
	Total	0.00	46.00	13.92	9.06
N		129			

NBQ: Neck Bournemouth Questionnaire; NDI: Neck Disability Index.

**Table 4 healthcare-11-01926-t004:** KMO and Bartlett´s test of NBQ-Sp.

Kaiser–Meyer–Olkin Measure of Sampling Adequacy		0.857
Bartlett’s test of sphericity	Approx. Chi-squared	566.868
	gl	21
	Sig.	<0.001

**Table 5 healthcare-11-01926-t005:** Total variance explained of NBQ-Sp.

Component	Initial Eigenvalues			Sums of Extraction of Charges Squared		
	Total	% of Variance	% Accumulated	Total	% of Variance	% Accumulated
1	4.398	62.834	62.834	4.398	62.834	62.834
2	0.815	11.641	74.475			
3	0.651	9.302	83.777			
4	0.415	5.935	89.712			
5	0.333	4.753	94.465			
6	0.261	3.723	98.188			
7	0.127	1.812	100.000			

**Table 6 healthcare-11-01926-t006:** Cronbach´s alpha and reliability of the NBQ-Sp.

	Total
Cronbach’s α	0.897
ICC (item responses)	0.866 [ICC 95%: 0.791–0.933]

**Table 7 healthcare-11-01926-t007:** Correlations between NBQ-Sp total scores with SF-12, NDI and EuroQol 5D y VAS questionnaires.

NBQ Total Scores		
SF-12	Physical Function	0.974 **
	Role—Physical	0.937 **
	Bodily Pain	0.972 **
	General Health	0.971 **
	Vitality	0.933 **
	Social Functioning	0.950 **
	Role—Emotional	0.938 **
	Mental Health	0.901 **
	Physical Component State	0.978 **
	Mental Component state	0.960 **
NDI Total	Pain Intensity	0.565 **
	Personal care	0.437 **
	Lifting weights	0.457 **
	Reading	0.482**
	Neck pain	0.447 **
	Concentrate on something	0.465 **
	Work and usual activities	0.500 **
	Driving	0.494 **
	Sleep	0.563 **
	Free-time activities	0.560 **
	NDI Total score	0.657 **
EuroQol_5D		−0.771 **
EuroQol_VAS		−0.564 **

** means *p* ≤ 0.001.

## Data Availability

Data is contained within the article or Appendix A.

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
