# Peer review of "Spanish Cross-Cultural Adaptation and Validation of Neck Bournemouth Questionnaire (NBQ) for Neck Pain Patients"

_healthcare, 2023, doi:10.3390/healthcare11131926_

Round 1

Reviewer 1 Report

healthcare-2406218

Thank you very much for inviting me to revier the paper entitled: “Spanish Cross-Cultural Adaptation and Validation of Neck Bournemounth Questionnaire for Neck Pain Patients”

General comments: interesting manuscript that addresses an important área of medical need. Instruments that measeure the factors involved in neck pain accurately are needed for Clinical assessment and follow-up.  The Neck Bournemouth Questionnaire (NBQ) assesses various health domains, such as pain, function, disability, psychological and social aspects of patients with cervical pathologies

Please see may specific comments below for more details.

This manuscript consists of a non-structured abstract with 5 keywords, 5 sections (introduction, methodology with 8 subsections (subsection 2.6 with 3 sections), results, discussion with 7 subsections, and conclusions) on 13 pages of single-spaced text with embedded figures and tables. There are 35 references,2 figure and 6 tables. And as supplementary material the NBQ questionnaire in Spanish.

 Specific comments:

As a recommendation: Title: I would change the order and add the acronym of the questionnaire “Neck Bournemounth Questionnaire for Neck Pain Patients (NBQ) an Spanish Cross-Cultural Adaptation and Validation”

1.     The keywords are absolutely fine.

2.     Background: complete, well strutuctured.

3.     Introduction: Complete, well structured and with current references.

4.     Methodology: the methodology is current, Figure 1 complete and easy to read. The Questionnaires for construct validity are Quality of Life SF-12; EuroQoL 5-d/VAS, Neck Disability Index… I consider them relevant and valid.

5.     Results: I consider Table 3 needs correction, please.

6.     Discussion: subsection 4.1. Translation and cross-cultural adaptation: line 4 (I'm sorry but not having put the template with the number of lines is more difficult to specify): “of the available literature24 were followed” I understand that "literature24" is a "literature24"

I agree with the authors that it is an easy-to-understand questionnaire for the population and quick to administer both in the clinical and research settings, which facilitates its use by professionals.

Thanks again for the invitation.

Author Response

Dear reviewer,

We would like to thank the Editor and reviewers for their thoughtful and constructive comments. We have considered all suggestions, and have incorporated them into the revised manuscript. Changes to the original manuscript are identified by highlights (in yellow background). After corrections made, we believe that our document is much easier to read and understand. An itemized point-by-point response to the reviewers’ comments is presented below. 

Thank you very much for offering us the possibility of reviewing the document and being able to complement it with the suggestions and comments made by the reviewers. We have followed all the suggestions made by the reviewer to understand that the document evolves positively.

Reviewer: 1

As a recommendation: Title: I would change the order and add the acronym of the questionnaire “Neck Bournemounth Questionnaire for Neck Pain Patients (NBQ) an Spanish Cross-Cultural Adaptation and Validation”

Authors’ answer: Thank you for your suggestion. The order was change and the acronym was added.

  1. The keywords are absolutely fine.

Authors’ answer: Thank you for your appreciation.

  1. Background: complete, well structured.

Authors’ answer: Thank you for your recognition.

  1. Introduction: Complete, well structured and with current references.

Authors’ answer: Thank you for your appreciation.

  1. Methodology: the methodology is current, Figure 1 complete and easy to read. The Questionnaires for construct validity are Quality of Life SF-12; EuroQoL 5-d/VAS, Neck Disability Index… I consider them relevant and valid.

Authors’ answer: Thank you for your recognition.

  1. Results: I consider Table 3 needs correction, please.

Authors’ answer: Thank you for your comment, table 3 has been corrected.

  1. Discussion: subsection 4.1. Translation and cross-cultural adaptation: line 4 (I'm sorry but not having put the template with the number of lines is more difficult to specify): “of the available literature24 were followed” I understand that "literature24" is a "literature24"

Authors’ answer. Thank you for your appreciation. This is a grammatical error which has been corrected.

I agree with the authors that it is an easy-to-understand questionnaire for the population and quick to administer both in the clinical and research settings, which facilitates its use by professionals.

Authors’ answer: Thank you for your words.

Reviewer 2 Report

Dear authors.

In this manuscript, the cross-cultural adaptation and validation for the Spanish population of a questionnaire for the assessment of patients with neck pain is performed for the first time.

After reviewing the manuscript, I have to make the following comments and suggestions:

- The abstract gives an appropriate overview of the research, although I consider that it would lack to include what the objective of this research is.

- The introduction of this article is well elaborated and adequately addresses the existing problems regarding the research topic and the importance of adapting and validating this questionnaire to the Spanish population. I consider that the acronym PROM should be explained as this is the first time it is used in the main text of the manuscript, although it has already been specified in the abstract.

- The stated objective is adequate to clarify the study problem posed, although I consider that it should be written at the end of the introduction and not at the beginning of the discussion as it is in the current manuscript.

- The methodology allows the authors to adequately and comprehensively address the study problem in order to achieve the proposed objective. I believe that the inclusion criteria should clarify what is meant by recurrent neck pain. Normally this definition refers to episodes of pain separated by 3-month intervals, although the results refer to acute, subacute and chronic pain depending on the time of duration.

- The results have been presented in a clear manner to facilitate their understanding, although I consider that a table with the items of the questionnaire as they will be asked to the patients after this adaptation and validation should be included in the manuscript, and not as accompanying information.

- The discussion provides a detailed and in-depth analysis of the results obtained, establishing relationships with previous existing studies on the subject of this research.

- The conclusion is consistent with the results obtained, responding adequately to the proposed objective and the currently existing scientific evidence.

- The references are appropriate to address this topic of study and most of them are up to date.

 Kind regards.

Author Response

Dear reviewer,

We would like to thank the Editor and reviewers for their thoughtful and constructive comments. We have considered all suggestions, and have incorporated them into the revised manuscript. Changes to the original manuscript are identified by highlights (in yellow background). After corrections made, we believe that our document is much easier to read and understand. An itemized point-by-point response to the reviewers’ comments is presented below. 

Thank you very much for offering us the possibility of reviewing the document and being able to complement it with the suggestions and comments made by the reviewers. We have followed all the suggestions made by the reviewer to understand that the document evolves positively.

Reviewer: 2

In this manuscript, the cross-cultural adaptation and validation for the Spanish population of a questionnaire for the assessment of patients with neck pain is performed for the first time.

After reviewing the manuscript, I have to make the following comments and suggestions:

- The abstract gives an appropriate overview of the research, although I consider that it would lack to include what the objective of this research is.

Authors’ answer: Thank you for your comment.  The objective of the study has been added to the abstract.

- The introduction of this article is well elaborated and adequately addresses the existing problems regarding the research topic and the importance of adapting and validating this questionnaire to the Spanish population. I consider that the acronym PROM should be explained as this is the first time it is used in the main text of the manuscript, although it has already been specified in the abstract.

Authors’ answer: Thank you for your comment.

- The stated objective is adequate to clarify the study problem posed, although I consider that it should be written at the end of the introduction and not at the beginning of the discussion as it is in the current manuscript.

Authors’ answer: Thank you for your comment. The objective is now at the end of the introduction.

- The methodology allows the authors to adequately and comprehensively address the study problem in order to achieve the proposed objective. I believe that the inclusion criteria should clarify what is meant by recurrent neck pain. Normally this definition refers to episodes of pain separated by 3-month intervals, although the results refer to acute, subacute and chronic pain depending on the time of duration.

Authors’ answer: Thank you for your comment.  The term "recurrent" has been changed in order to avoid confusion. The correction can be found in subsection 2.1.participants of the Methodology section.

- The results have been presented in a clear manner to facilitate their understanding, although I consider that a table with the items of the questionnaire as they will be asked to the patients after this adaptation and validation should be included in the manuscript, and not as accompanying information.

Authors’ answer: Thank you for your comment.  A table with the NBQ-Sp items has been added as Table 1.

- The discussion provides a detailed and in-depth analysis of the results obtained, establishing relationships with previous existing studies on the subject of this research.

Authors’ answer: Thank you for your appreciation.

- The conclusion is consistent with the results obtained, responding adequately to the proposed objective and the currently existing scientific evidence.

Authors’ answer: Thank you for your recognition.

- The references are appropriate to address this topic of study and most of them are up to date.

Authors’ answer: Thank you for your appreciation.

Reviewer 3 Report

The objective of the manuscript (ID: healthcare-2406218) was to evaluate the validity and reliability of the Neck Bournemouth Questionnaire (NBQ) as a tool for clinicians and researchers to measure neck pain in the Spanish population.   

In accordance with the objective of this manuscript, the Introduction section provides a comprehensive overview of the magnitude of the problem that neck pain represents around the world, using adequate indicators of the frequency and burden of the disease on society. Also, the most frequently used questionnaires for the assessment of neck pain are presented in detail. In addition, the need to carry out translation and cross-cultural adaptation of the NBQ into Spanish (NBQ-Sp) and validation of the NBQ-Sp was clearly indicated. In accordance with the above, at the end of the Introduction section, the goals of this study are defined.

In the section Methodology are specified Study design, criteria for inclusion and exclusion of respondents in the study, Study sample size calculation (with citation of the appropriate reference; the data is additionally confirmed in subsection Data analysis, on page 5/6), Questionnaire, with Translation/adaptation according to internationally accepted methodology. In addition, 3 tools are listed that were applied in order to evaluate the construct validity of NBO-Sp (with citation of the appropriate references). In subsection Data analysis, a detailed description of all statistical procedures applied in this study is presented.

In the Results section, the results, which are in accordance with the defined goals in this study, are clearly and clearly presented.

Significant corrections should be made in the Discussion section. Subsection `Strengths and weaknesses' contains a fair discussion of the strengths and weaknesses of this study, with a proposal for overcoming the weaknesses.

Section Conclusion is consistent with the results of this study.   

Some corrections are necessary (major revision):

  • In the Abstract:
    • Clearly state the goal of the work;
    • Enter the meaning of abbreviations (ICC, SEM, MDC);
    • Check and correct the accuracy of the part of the sentence that reads `... the Bartlett's test was p<0.001 (p<0.05), ...`. Explain or correct.
  • Introduction: Explain the abbreviation `Neck pain (CP)'.
  • Methodology:
    • Add subsection `Study setting` and subsection `Study population`. Describe how participants were recruited for this study. Specify the type of `Study sample'.
    • State that all questionnaires used in this study were self-reported.
  • Results:
    • In Table 1, add data for the distribution of respondents by age. 
  • Discussion:
    • Avoid citing Figure and Table in the Discussion section;
    • Some validation studies have reported a two-factor structure for the NBO, in contrast to the original and Spanish versions. Give a possible explanation for the mentioned differences;
    • Avoid unnecessary and excessive repetition of text from previous sections (Introduction, Methodology); Pay attention to repeating and quoting references that have already been mentioned in sections Introduction and Methodology; 
    • Only 1 reference was introduced in the entire Discussion section, which was not cited in the Introduction and Methodology sections (it is reference No 35). Explain and correct.

The quality of English linguage is appropriate.  

Author Response

Dear reviewer,

We would like to thank the Editor and reviewers for their thoughtful and constructive comments. We have considered all suggestions, and have incorporated them into the revised manuscript. Changes to the original manuscript are identified by highlights (in yellow background). After corrections made, we believe that our document is much easier to read and understand. An itemized point-by-point response to the reviewers’ comments is presented below. 

Thank you very much for offering us the possibility of reviewing the document and being able to complement it with the suggestions and comments made by the reviewers. We have followed all the suggestions made by the reviewer to understand that the document evolves positively.

Reviewer: 3

The objective of the manuscript (ID: healthcare-2406218) was to evaluate the validity and reliability of the Neck Bournemouth Questionnaire (NBQ) as a tool for clinicians and researchers to measure neck pain in the Spanish population.  

In accordance with the objective of this manuscript, the Introduction section provides a comprehensive overview of the magnitude of the problem that neck pain represents around the world, using adequate indicators of the frequency and burden of the disease on society. Also, the most frequently used questionnaires for the assessment of neck pain are presented in detail. In addition, the need to carry out translation and cross-cultural adaptation of the NBQ into Spanish (NBQ-Sp) and validation of the NBQ-Sp was clearly indicated. In accordance with the above, at the end of the Introduction section, the goals of this study are defined.

Authors’ answer: Thank you for your appreciation.

In the section Methodology are specified Study design, criteria for inclusion and exclusion of respondents in the study, Study sample size calculation (with citation of the appropriate reference; the data is additionally confirmed in subsection Data analysis, on page 5/6), Questionnaire, with Translation/adaptation according to internationally accepted methodology. In addition, 3 tools are listed that were applied in order to evaluate the construct validity of NBO-Sp (with citation of the appropriate references). In subsection Data analysis, a detailed description of all statistical procedures applied in this study is presented.

Authors’ answer: Thank you for your recognition.  

In the Results section, the results, which are in accordance with the defined goals in this study, are clearly and clearly presented.

Authors’ answer: Thank you for your appreciation.

Significant corrections should be made in the Discussion section. Subsection `Strengths and weaknesses' contains a fair discussion of the strengths and weaknesses of this study, with a proposal for overcoming the weaknesses.

Authors’ answer: Thank you for your comment.

Section Conclusion is consistent with the results of this study.  

Authors’ answer: Thank you for your comment.

Some corrections are necessary (major revision):

In the Abstract:

Clearly state the goal of the work;

Authors’ answer: Thank you for your suggestion.  The objective is now in the abstract.

Enter the meaning of abbreviations (ICC, SEM, MDC);

Authors’ answer: Thank you for your comment. The meaning of the acronym has been added.

Check and correct the accuracy of the part of the sentence that reads `... the Bartlett's test was p<0.001 (p<0.05), ...`. Explain or correct.

Authors’ answer: Thank you for your comment. It is now corrected.

Introduction: Explain the abbreviation `Neck pain (CP)'.

Authors’ answer: Thank you for your comment. It´s now corrected, we wanted to write NP (Neck Pain) instead of CP (Cervical Pain).

Methodology:

Add subsection `Study setting` and subsection `Study population`. Describe how participants were recruited for this study. Specify the type of `Study sample'.

Authors’ answer: Thank you for your comment.  Those subsections are already in the methodology section, describing how participants were recruited (sub-section Study setting) as well as the type of study sample that was used (at the end of the sub-section Study population).

State that all questionnaires used in this study were self-reported.

Authors’ answer: Thank you for your comment. It´s now added in the section “data collection” of the methodology.

Results:

In Table 1, add data for the distribution of respondents by age.

Authors’ answer: Thank you for your comment. The distribution of respondents by age is now in that table.

Discussion:

Avoid citing Figure and Table in the Discussion section;

Authors’ answer: Thank you for your comment.  These quotations have been removed from the discussion section.

Some validation studies have reported a two-factor structure for the NBO, in contrast to the original and Spanish versions. Give a possible explanation for the mentioned differences;

Authors’ answer: Thank you for your comment. The differences in the structure of the NBQ with respect to other previously published versions may be due to socio-cultural differences between the two population groups. These differences could determine the different weights of each of the items in each of the factors, which would condition this difference. Even within the same culture, it is sometimes necessary to carry out validation analyses in different population groups, precisely in order to be able to evaluate possible differences between the groups, and there are many examples in the scientific literature.

Avoid unnecessary and excessive repetition of text from previous sections (Introduction, Methodology); Pay attention to repeating and quoting references that have already been mentioned in sections Introduction and Methodology;

Authors’ answer: Thank you for your comment. The discussion section has been revised to avoid excessive mention of content from previous sections.

Only 1 reference was introduced in the entire Discussion section, which was not cited in the Introduction and Methodology sections (it is reference No 35). Explain and correct.

Authors’ answer: Thank you for your comment. The references used in the discussion section were added to the introduction section for information purposes only. This citation error has now been corrected and the references now only appear in the discussion section.

Round 2

Reviewer 3 Report

The revised manuscript (ID: healthcare-2406218) provides important and relevant information about a results validity and reliability of the Spanish version of the of Neck Bournemouth Questionnaire for neck pain patients (NBQ-Sp). 

The authors addressed all my comments in a comprehensive manner. 

I hope that all the corrections made in the Abstract, Methods, Results and Discussion sections contributed to making the paper more transparent for the readers. 

I applaud the authors for their napor to identify characteristics of NBQ-Sp, which will most importantly lead researchers to assess neck pain in the Spanish-speaking population.   

The quality of English linguage is appropriate. 

Author Response

RESPONSE TO REVIWER: Itemized List

HEALTHCARE

Manuscript ID

healthcare-2406218

We would like to thank the Editor and reviewers for their thoughtful and constructive comments.

Reviewer 3

The revised manuscript (ID: healthcare-2406218) provides important and relevant information about a results validity and reliability of the Spanish version of the of Neck Bournemouth Questionnaire for neck pain patients (NBQ-Sp).

The authors addressed all my comments in a comprehensive manner.

I hope that all the corrections made in the Abstract, Methods, Results and Discussion sections contributed to making the paper more transparent for the readers.

I applaud the authors for their napor to identify characteristics of NBQ-Sp, which will most importantly lead researchers to assess neck pain in the Spanish-speaking population.

Author´s Answer: The authors are grateful for all comments and suggestions for improvement of the manuscript.
